# The interdependent network of gene regulation and metabolism is robust where it needs to be

David F. Klosik[1], Anne Grimbs [2], Stefan Bornholdt[1] & Marc-Thorsten Hütt[2]

Despite being highly interdependent, the major biochemical networks of the living cell—the networks of interacting genes and of metabolic reactions, respectively—have been approached mostly as separate systems so far. Recently, a framework for interdependent networks has emerged in the context of statistical physics. In a first quantitative application of this framework to systems biology, here we study the interdependent network of gene regulation and metabolism for the model organism *Escherichia coli* in terms of a biologically motivated percolation model. Particularly, we approach the system's conflicting tasks of reacting rapidly to (internal and external) perturbations, while being robust to minor environmental fluctuations. Considering its response to perturbations that are localized with respect to functional criteria, we find the interdependent system to be sensitive to gene regulatory and protein-level perturbations, yet robust against metabolic changes. We expect this approach to be applicable to a range of other interdependent networks.

[1] Institute for Theoretical Physics, University of Bremen, Hochschulring 18, 28359 Bremen, Germany. [2] Department of Life Sciences and Chemistry, Jacobs University, Campus Ring 1, 28759 Bremen, Germany. Correspondence and requests for materials should be addressed to S.B. (email: bornholdt@itp.uni-bremen.de) or to M.-T.H. (email: m.huett@jacobs-university.de)

A main conceptual approach of current research in the life sciences is to advance from a detailed analysis of individual molecular components and processes towards a description of biological systems and to understand the emergence of biological function from the interdependencies on the molecular level. Supported by the diverse high-throughput 'omics' technologies, the relatively recent discipline of systems biology has been the major driving force behind this new perspective, which becomes manifest, for example, in the effort to compile extensive databases of biological information to be used in genome-scale models[1–3]. Despite its holistic 'game plan', however, systems biology frequently operates on the level of subsystems: Even when considering cell-wide transcriptional regulatory networks, as, e.g., in network motif analysis[4], this is only one of the cell's networks. Likewise, the popular approach to studying metabolic networks in systems biology, constraint-based modelling, accounts for steady-state predictions of metabolic fluxes of genome-scale metabolic networks[5], which again, is only one of the other networks of the cell.

In the analysis of such large networks, systems biology draws its tools considerably from the science of complex networks, which, by combining the mathematical subdiscipline of graph theory with methods from statistical physics, greatly contributed to the understanding of, e.g., the percolation properties of networks[6], potential processes of network formation[7] or the spreading of disease on networks[8]. In the early 2000s, gene regulation and metabolism have been among the first applications of 'network biology'[9]. The most prominent findings on the gene regulatory side concerned the statistical observation and functional interpretation of small over-represented subgraphs (network motifs)[10, 11] and the hierarchical organization of gene regulatory networks[12]. On the metabolic side, the broad degree distribution of metabolic networks stands out[13], with the caveat, however, that 'currency metabolites' (as ATP and $H_2O$) can severely affect network properties[14], as well as the hierarchical modular organization of metabolic networks[15, 16].

Recently, the field of complex networks moved its focus from single networks to the interplay of networks that interact with and/or depend on each other (multilayer networks, networks of networks). Strikingly, it turned out that explicit interdependency between network constituents can fundamentally alter the percolation properties of the resulting interdependent networks, which can show a discontinuous percolation transition in contrast to the continuous behavior in single-network percolation[17–23]. Also, contrary to the isolated-network case, networks with broader degree distribution become remarkably fragile as interdependent networks[17].

However, this set of recent developments in network science about fragility due to interdependence still lacks application to systems biology. In Reis et al.[24] the question of robustness in multilayer biological networks has been raised for the first time (see also Bianconi[25]). Specifically, it has been shown how specific correlations of the intra- and interlayer connections reduce cascading failures and thus provide a robust multilayer network architecture. Relevant progress in the application of concepts of multilayer networks has been made, for instance, in transportation infrastructures[26] and brain networks[24]. In such applications, the discovery of new mechanisms went along with advances in the theoretical foundation in exploring dynamical processes in multilayer networks, for example, diffusion processes[27], spreading processes[28] and message passing[29–31].

Arguably, the most prominent representative of interdependent networks in a biological cell is the combined system of gene regulation and metabolism, which are interconnected by various forms of protein interactions, e.g., enzyme catalysis of biochemical reactions couples the regulatory to metabolic network, while the activation or deactivation of transcription factors by certain metabolic compounds provides a coupling in the opposite direction.

Although it is well-known that gene regulatory and metabolic processes are highly dependent on one another, only few studies addressed their interplay on a larger, systematic scale[32–34]. The first two studies (Covert et al.[32] and Shlomi et al.[33]) aimed at finding consistent metabolic-regulatory steady states by translating the influence of metabolic processes on gene activity into metabolic flux predicates and incorporating high-throughput gene expression data. This can be considered as an extension of the constraints-based modelling framework beyond the metabolic subsystem. In the paper of Samal and Jain[34], on the other hand, the transcriptional regulatory network of *Escherichia coli* (*E. coli*) metabolism has been studied as a Boolean network model with flux predicates represented by additional interactions.

The formalism of interdependent networks now allows us to go beyond these important pioneering works on integrative models, by analysing the robustness of the combined system in terms of the maximal effect of a small perturbation. In particular, the findings can be interpreted in the context of cascading failures and percolation theory.

We here formulate a first application of the new methodological perspective to the combined networks of gene regulation and metabolism in *E. coli*. Using various biological databases, particularly EcoCyc[35, 36] as the main core, we have compiled a graph representation of gene regulatory and metabolic processes of *E. coli*, including a high level of detail in the structural description, distinguishing between a comparatively large number of vertex and edge types according to their biological functionality.

A structural analysis of this compilation reveals that, in addition to a small set of direct edges, the gene regulatory and the metabolic domains are predominantly coupled via a third network domain consisting of proteins and their interactions. This rich structural description, together with a purpose-built, biologically motivated percolation model allows us to assess the functional level with methods derived from percolation theory. More precisely, we investigate cascading failures in the three-domain system, emanating from small perturbations, localized in one of the domains. We observe below that (i) randomized versions of the graph are much less robust than the original graph and (ii) that the integrative system is much more susceptible to small perturbations in the gene regulatory domain than in the metabolic one.

## Results

The core object of our investigation is an *E. coli* network representation of its combined gene regulation and metabolism, which can be thought of as functionally divided into three domains. Aside from the obvious domains emerging from gene regulatory and metabolic processes, these processes are connected by an intermediate domain that models both, the enzymatic influence of genes on metabolic processes, as well as signalling-effects of the metabolism on the activation or inhibition of the expression of certain genes. Accordingly, the underlying interaction graph $G = (V, E) = \{G_R, G_I, G_M\}$ with its set of vertices $V$ and edges $E$ consists of three interconnected subgraphs, the gene regulatory domain $G_R$, the interface domain $G_I$ and the metabolic domain $G_M$. A sketch of the network model and some of its properties are presented in Fig. 1. Further information and a detailed characterization are given in the "Methods" section (as well as in A.G., D.F.K., S.B., and M.-T.H., manuscript in preparation).

Of central importance for modelling spreading dynamics in our system are the functionally different roles of the edges in the

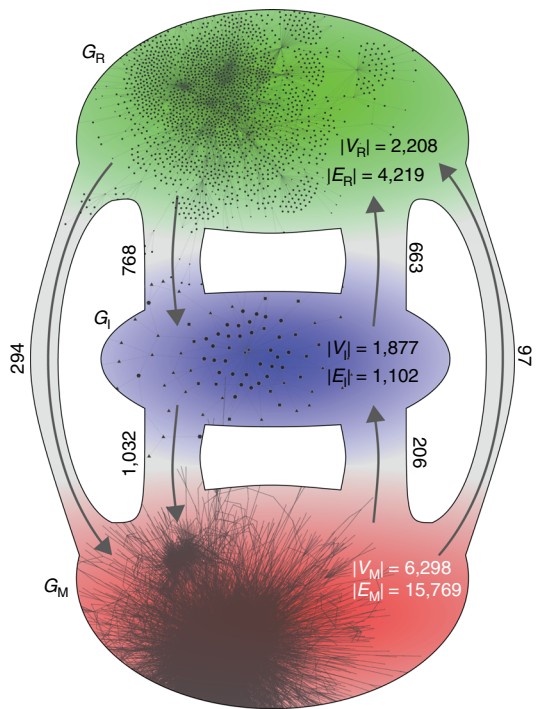

**Fig. 1** Sketch of the domain organization of the integrative *E. coli* network. The gene regulatory domain, $G_R$, (*top*) is predominantly connected to the metabolic domain, $G_M$, (*bottom*) via a protein-interface layer, $G_I$. For each domain, the number of vertices, |V|, and the number of edges, |E|, are given as well as the number of inter-domain edges (attached to the *arrows*). For illustrative purposes, snapshots of the largest weakly-connected components of $G_R$, $G_I$ and $G_M$ are included

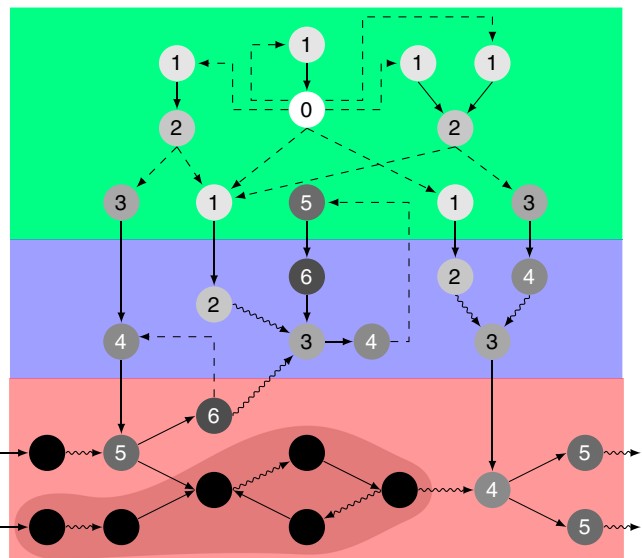

**Fig. 2** Sketch of the propagation dynamics of a single-vertex perturbation. The perturbation in the small sample three-domain organized network with gene regulatory domain (*top*, green), (protein-)interface (*middle*, blue) and metabolic domain (*bottom*, red) is initiated in vertex 0 and spreads according to the percolation model given in Eqs. (1) and (2). The numbers and the grayscale indicate the time step in which the respective vertex has been turned off; *black vertices* are not affected. The *arrow line* styles denote the three logical categories of edges: C: *curly*, D: *solid*, and R: *dashed*. In the subsequent steps of the analysis, we consider the largest weakly connected component of the frozen network state (vertices in the *dark area*). A more detailed illustration is given in Supplementary Fig. 7

system, which we capture in three logical categories of an edge (LCE):

C, 'conjunct': The target vertex of an edge with this logical AND property depends on the source vertex, i.e., it will fail once the source vertex fails. For example, for a reaction to take place, all of its educts have to be available.

D, 'disjunct': Edges with this logical OR property are considered redundant in the sense that a vertex only fails if the source vertices of all of its incoming D-edges fail. For instance, a compound will only become unavailable once all of its producing reactions have been canceled.

R, 'regulation': Edges of this category cover 14 different kinds of regulatory events (described in detail in Supplementary Note 2). As shown below, in terms of the propagation dynamics we treat these edges similar to the 'conjunct' ones.

As outlined in the "Methods" section the conjunct and disjunct edge properties in our study have been derived from the underlying biological mechanisms. The disjunct edges may be examples of the redundant interdependencies, which have been discussed in Radicchi and Bianconi[31] as a possible foundation of the robustness of multilayer (specifically, multiplex) networks.

In order to assess the effect of perturbations in this three-domain organized system, we devise a biologically motivated percolation model that determines the sets of potentially affected vertices (see "Methods" section). An illustration of the dynamics is shown in Fig. 2.

As a side remark, the spreading of a perturbation according to the rules defined above could also be considered as an epidemic process with one set of connections with a very large, and a second set of connections with a very low probability of infection[37].

**Comparison with previous approaches**. In systems that can be described without explicit dependencies between its constituents, but with a notion of functionality that coincides with connectivity, percolation theory is a method of first choice to investigate the system's response to perturbations of a given size that can be modelled as failing vertices or edges[6, 38]. The fractional size of the giant connected component as a function of the occupation probability $p$ of a constituent typically vanishes at some critical value $p_c$, the percolation threshold. In the following, we will mostly use the complementary quantity $q = 1 - p$ so that $q_c = 1 - p_c$ can be interpreted as the critical size of the initial attack or perturbation of the system. The strong fluctuations of the system's responses in the vicinity of this point can serve as a proxy for the percolation threshold, which is especially useful in finite systems in which the transition appears smoothed out. In our analysis, the susceptibility $\tilde{\chi} = \langle S^2 \rangle - \langle S \rangle^2$, where $S$ is the size of the largest cluster, is used[39].

Upon the introduction of explicit dependencies between the system's constituents, the percolation properties can change dramatically. The order parameter no longer vanishes continuously but typically jumps at $p_c$ in a discontinuous transition[18, 19] as cascades of failures fragment the system. A broader degree distribution now enhances a graph's vulnerability to random failures, in opposition to the behavior in isolated graphs[17, 40]. Details of the corresponding theoretical framework have been worked out by Parshani et al.[18], Son et al.[20], Baxter et al.[41] and more recently the notions of 'networks of networks'[42–45] and network recovery[46, 47] have been included. There have also been attempts to capture this class of models in a general framework of multilayer networks[48]. Important contributions to the theoretical understanding of percolation phenomena in multilayered structures come from studying variants of message passing algorithms on multiplex networks. Cellai et al.[29], for example, show how the

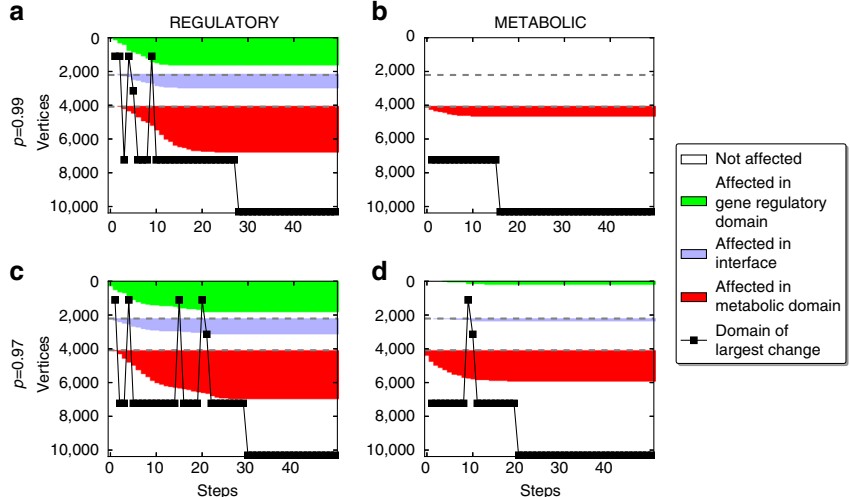

**Fig. 3** Sample trajectories of the integrative *E. coli* network. Initial perturbations differ in size, $q = 1 - p = 0.01$ (**a**, **b**) and $q = 0.03$ (**c**, **d**), as well as in localization: perturbations are initiated in the gene regulatory domain (**a**, **c**) and the metabolic domain (**b**, **d**). In each panel, the vertex states are plotted against the update steps according to Eqs. (1) and (2). The vertices are grouped according to the domain affiliation, starting from gene regulatory domain (*top*, *green*) via protein-interface domain (*middle*, *blue*) towards metabolic domain (*bottom*, *red*). Within each domain they are arranged with respect to the update step, in which they were affected in the respective run (vertex ordering within the domains is thus different from run to run). Affected vertices are shown in the color of their domain, unaffected vertices are shown last for each run and in *white*. The *horizontal dashed lines* denote the domain boundaries and the *black square* indicates the domain with the largest number of newly affected vertices in each step. Connections between *black squares* are only shown to guide the eye

presence of link overlap between layers can be considered in terms of two similar, but conceptually different, processes and analyse the behavior of the corresponding order parameters.

In addition to random vertex failure other procedures for initial vertex removal have been explored, e.g., vertex removal with respect to their degree (targeted attacks)[49] or localized attacks, for which currently two notions have been described. Localized attacks of the first sort are defined on spatially embedded networks and are 'local' with respect to a distance in this embedding, i.e., in a 'geographical' sense[50, 51]. The second approach considers locality in terms of connectivity: around a randomly chosen root vertex, neighbors are removed layer by layer[52, 53]. In contrast, as described below in our approach, attacks are localized with respect to the three network domains, while within the domains vertices are chosen randomly.

At this point, we would like to shortly comment on the applicability of the mathematical concepts of interdependent networks to real-world data. Clear conceptual categories such as the distinction between dependency edges and connectivity edges have been instrumental in gaining theoretical insight into the properties of interdendent systems. Similarly, in the theoretical work on multilayer networks (see, e.g., De Domenico et al.[54], Boccaletti et al.[55]), it is assumed that the assignment of vertices and edges to layers is defined a priori. It is worth mentioning, however, that in order to adapt these frameworks to the situation at hand several of these categories require adaptation. We envision that for many real-life applications a certain diversity of vertex and edge attributes will be required and the multilayer structure will rather emerge from the arrangement of such different edge types among the different vertex types in the system. Effectively, some classes of edges may then represent simple connectivity, while others can rather be seen as dependency edges. In biology, such dependencies are typically mediated by specific molecules (e.g., a small metabolite affecting a transcription factor, or a gene encoding an enzyme catalysing a biochemical reaction). Such implementations of dependency edges are no longer just associations and it is hard to formally distinguish them from the functional edges.

With the explicitly alternating 'percolation' and 'dependency' steps in typical computational models of the kind originally introduced in Buldyrev et al.[17] in mind we would like to point out that in our directed model edges of all types are updated in every time step. If the dependencies of a vertex are fulfilled (here: only intact C- and R-neighbors), it will only fail once it loses all connectivity (here: via D-edges), irrespective of the current component structure, which is only considered after the dynamics is frozen.

**Network response to localized perturbations**. The main feature of our reconstructed network, the three-domain structure based on the biological role of its constituents, allows us to study the influence of localizing the initial perturbation (see "Methods" section). Thus, although we will not focus on (topological) details of the graph here (which will be presented in (A.G., D.F.K., S.B., and M.-T.H., manuscript in preparation)), already from the vertex and edge counts in Fig. 1, we see that the domains are of different structure. While the regulatory and the metabolic subgraphs, $G_R$ and $G_M$ have average (internal) degrees of about 1.9 and 2.5, the interface subgraph, $G_I$, is very sparse with $\langle k \rangle \approx 0.6$ and we can expect it to be fragmented. Hence, in the following, we decide to only perturb in $G_R$ and $G_M$.

In a first step, we sample some full cascade trajectories in order to check our expectation of different responses of the system to small perturbations applied in either $G_R$ or $G_M$; two rather large values of $q$ are chosen and the raw number of unaffected vertices is logged during the cascade. Indeed, already this first approach implies a different robustness of the gene regulatory and the metabolic domains in terms of the transmission of perturbation cascades to the other domains. Cascades initiated in the metabolic domain of the network tend to be rather restricted to this domain, while the system seems much more susceptible to small perturbations applied in the gene regulatory domain. This effect can be seen both in the overall sizes of the aggregated cascades as well as in the domain which shows the largest change with respect to the previous time step (largest set of newly affected vertices), which we indicate by *black markers* in Fig. 3. More sample

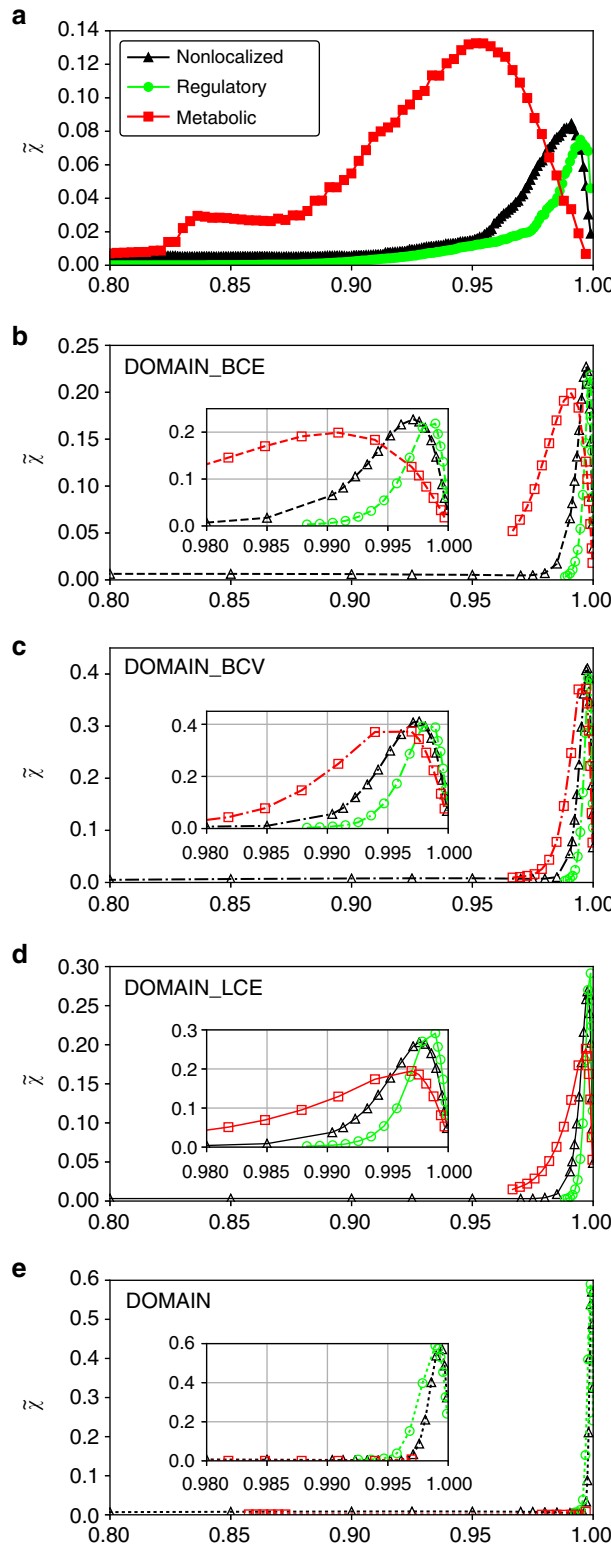

**Fig. 4** Susceptibility of the integrative *E. coli* network and its randomized versions for distinct localized perturbations. In each panel, the average over 500 runs for an initial perturbation size, $q = 1 − p$, of the original network **a** and the four randomization schemes in decreasing stringency order **b**–**e** is shown. For each scheme 500 graph realizations have been considered. While non-localized perturbations are presented in *black*, perturbations initiated in gene regulatory domain and metabolic domain are given in *green* and in *red*, respectively

trajectories are shown in Supplementary Figs. 1–5 and, although they illustrate occasionally large fluctuations between the behaviors of single trajectories, they are consistent with this first observation. They also show that considerably larger metabolic perturbations are needed for large cascades and back-and-forth propagation between domains to emerge.

After this first glance at the system, we aim for a more systematic approach and apply our analysis as described above: we compute cascade steady-states $\sigma_\infty$ but now we choose the largest (weakly) connected component $B(X, q)$ as the order parameter and compute the susceptibility according to Eq. (3), the peak-position of which, when considered as a function of $q = 1 − p$, we use as a proxy for the perturbation size at which the interdependent system breaks down.

The results for different initially perturbed domains illustrate that, indeed, a considerably lower $p_c$ (i.e., larger critical perturbation size $q_c$) is estimated in the case of metabolic perturbations compared to regulatory or non-localized ones (Fig. 4a). For each point, we average 500 runs for the corresponding set of parameters.

In order to assess whether the above-described behavior is due to specific properties of the network, we use sets of randomized graphs. For each of the four randomization schemes, described in the "Methods" section, we prepared 500 graph instances and repeated the analysis for each of them as done before for the single original graph. The corresponding results for the susceptibility (Fig. 4b–e) yield two major observations: firstly, metabolic perturbations still lead to, albeit only slightly, higher $q_c = 1 − p_c$ estimates (with exception of DOMAIN randomization). Thus, the system's tendency to be more robust towards metabolic perturbations is largely preserved. Secondly, we see that overall the original network seems to be much more robust than the randomized networks; very small perturbations are sufficient to break the latter ones. The robustness of the original graph, thus, cannot be solely due to the edge and vertex properties kept in the randomization schemes.

Finally, let us focus on the practical aspect of these findings. Beyond the careful statistical analysis described above, a quantity of practical relevance is the average size of the unaffected part of the system under a perturbation of given size q. For this purpose, we examine the fractions of unaffected vertices, $A(X, q)$, after cascades emanating from perturbations of different sizes and initiated in different domains, regardless of the resulting component structure and for both, the original graph and the shuffled ones (Fig. 5).

The number of unaffected vertices for the real network is much larger than for all four randomization schemes, suggesting a strong overall robustness of the biological system. Distinguishing, however, between the metabolic and the gene regulatory components reveals that the metabolic part is substantially more robust than the regulatory part (for not too large initial perturbations, $p > 0.94$).

## Discussion

We investigated the spreading of perturbations through the three domains of a graph representation of the integrative system of *E. coli*'s gene regulation and metabolism. Our results quantify the resulting cascading failures as a function of size and localization of the initial perturbation.

Our findings show that the interdependent network of gene regulation and metabolism unites sensitivity and robustness by showing different magnitudes of damage dependent on the site of perturbation.

While the interdependent network of these two domains is in general much more robust than its randomized variants

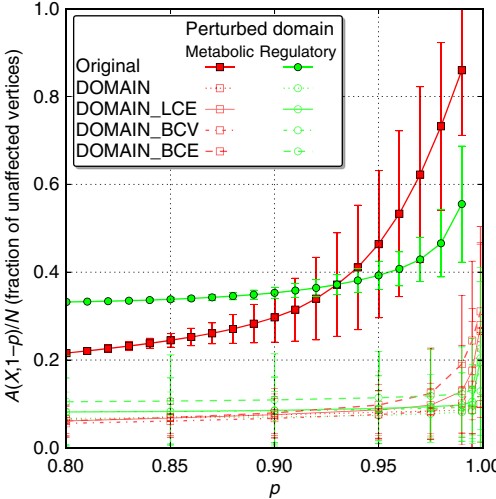

**Fig. 5** Fractions of unaffected vertices of the integrative *E. coli* network and its randomized versions for distinct localized perturbations. Per initial perturbation size, $q = 1 - p$, 5000 runs were averaged for the original network (*filled markers*) and the four randomization schemes (*open markers*), where in turn, 500 realizations are considered for each scheme. Initial perturbations in the gene regulatory domain are presented as *green dots*, while perturbations initiated in the metabolic domain are given as *red squares*. The *error bars* indicate s.d. (results for randomized realizations have been combined)

(retaining domain structure, degree sequence, and major biological aspects of the original system), a pronounced difference between the gene regulatory and metabolic domain is found: Small perturbations originating in the gene regulatory domain typically trigger far-reaching system-wide cascades, while small perturbations in the metabolic domain tend to remain more local and trigger much smaller cascades of perturbations.

In order to arrive at a more mechanistic understanding of this statistical observation, we estimated the percolation threshold of the system, $p_c$, and found that it is much lower (i.e., larger perturbations, $q_c = 1 - p_c$, are required) for perturbations initiated in the metabolic domain than for those applied to the gene regulatory domain.

This is in accordance with the intuition that the metabolic system is more directly coupled to the environment (via the uptake and secretion of metabolic compounds) than the gene regulatory domain. The distinct percolation thresholds therefore allow for implementing a functionally relevant balance between robustness and sensitivity: The biological system can achieve a robustness towards environmental changes, while—via the more sensitive gene regulatory domain—it still reacts flexibly to other systemic perturbations.

Discovering this design principle of the biological system required establishing a novel method of analysing the robustness of interdependent networks, the network response to localized perturbations: An interdependent network can have markedly different percolation thresholds, when probed with perturbations localized in one network component compared to another.

Lastly, we would like to emphasize that the application of the theoretical concepts of interdependent networks to real-life systems involves several non-trivial decisions:

In the vast majority of theoretical investigations, interdependent networks are defined via a distinction between dependency edges and connectivity-representing edges[17, 19, 56]. Often, in specific applications, this distinction of connectivity edges and dependency edges is not immediately clear (see

examples from neuroscience[57] and systems biology[58]). In Morone et al.[59], an elegant way of mapping such real-life systems to the formal description of connectivity edges and dependency edges has been worked out for the case of brain networks by emphasizing the different roles of intramodule and intermodule edges.

Here the consideration of the biological data with respect to different levels of description (an overview of the corresponding vertex and edge categories is given in Supplementary Note 2) results in a graph representation with a functional three-domain organization: Vertices involved in gene regulation, metabolic vertices, and vertices associated with the (protein) interface between these two main domains are interconnected with (functionally) different classes of edges. These edge classes are a key ingredient of our biologically motivated percolation model. As a consequence, our investigation required an adjustment of the original percolation model described in Buldyrev et al.[17] to the vertex and edge classes employed here. We expect that application of the notion of interdependent networks to real-life systems is likely to require adjustments of the percolation model, even though the main features of the original theory remain fully applicable.

As mentioned above, when dealing with (possibly incomplete) biological data, one major task is to find the right balance between radical simplifications of systemic descriptions and an appropriate level of detail, still allowing for a meaningful evaluation of the biological information, that is, to abstract from the minor but keep the essential details[60]. We have outlined that, in this study, we chose to incorporate high level of detail in the structural description, distinguishing between a comparatively large number of vertex and edge types. This rich structural description, which includes information about the functional relationships between vertices allows us to assess the dynamical/functional level with the comparatively simple methods derived from percolation theory.

Even though the framework of multiplex networks does not strictly apply to the example of a multilayer network discussed here, as there is no one-to-one mapping from vertices in one layer to vertices in the other layers, several theoretical findings on multiplex networks nevertheless can help us to put the results of our investigation in a broader context. It has been recognized in a previous study[31] that understanding the robustness of multilayer networks may require adapting the standard percolation model, where a vertex is switched off, if at least one of the interdependent vertices has been switched off as well. This percolation model is at the core of many of the investigations of catastrophic failures of interdependent networks, e.g., Buldyrev et al.[17]. In this scenario, a larger number of layers will typically lead to more vulnerable networks. The interesting question, how an increasing number of layers might actually enhance the robustness of the system, has been addressed in Radicchi and Bianconi[31]. In our case, the cascading failure, and hence the percolation model, is determined by the Boolean update rules in Eqs. (1) and (2), which have been derived from the underlying biological processes as described in the corresponding "Results" and "Methods" sections (and in more detail in the Supplementary Information).

This biologically motivated percolation model used here is comparable to the model of percolation in multiplex networks introduced in Radicchi and Bianconi[31]. Even though intracellular molecular network systems, as the one studied here, tend to not adhere to the formal requirement of a one-to-one mapping of vertices between different layers, it is likely that the main finding from[31] —that an addition of layers enhances systemic robustness by providing redundant interdependencies— may still hold. It would be interesting to further explore in what way the multilayer structure of intracellular molecular networks indeed enhances their robustness in this way.

Furthermore, an important question is, whether the analysis of the fragmentation of such a network under random removal of vertices can provide a reliable assessment of functional properties, since the response of such a molecular network clearly follows far more intricate dynamical rules than the percolation of perturbations can suggest.

A future step could include the construction of a Boolean network model for the full transcriptional regulatory network and the connection of this model to flux predictions obtained via flux balance analysis, a first attempt of which is given in Samal and Jain[34] (where the model of Covert et al.[32] with still fewer interdependence edges has been used).

Our perturbation spreading approach might help bridging the gap between theoretical concepts from statistical physics and biological data integration: Integrating diverse biological information into networks, estimating 'response patterns' to systemic perturbations and understanding the multiple systemic manifestations of perturbed, pathological states is perceived as the main challenge in systems medicine (see, e.g., Bauer et al.[61]). Concepts from statistical physics of complex networks may be of enormous importance for this line of research[62, 63].

While the simulation of the full dynamics is still problematic as our knowledge of the networks is still incomplete, our present strategy extracts first dynamical properties of the interdependent networks. At a later time point, we can expect qualitative advances from full dynamical simulations, however, dependent on the quality of the data sets.

On the theoretical side, future studies might shift the focus onto recasting the system into an appropriate spreading model, e.g., in the form of an unordered binary avalanche[64, 65], or as an instance of the Linear Threshold model[37] with a set of edges with a very high and a second set with a very low transmission probability (C/R- and D-edges, respectively).

Also, it would be interesting to investigate in more detail, whether the pattern of dependency edges, which in Reis et al.[24] has been associated with robustness of interdependent networks, are more prominent in the metabolism-to-regulation direction of our system than in the regulation-to-metabolism direction.

Radicchi[23] presents an approach for the investigation of the percolation properties of finite size interdependent networks with a specific adjacency matrix with the goal of loosening some of the assumptions underlying the usual models (e.g., infinite system limit, graphs as instances of network model). While this formalism allows for the investigation of many real-world systems, there are still restrictions as to the possible level of detail. In our special case, for instance, a considerable amount of information would be lost if the system was restricted to vertices with connections in both the C/R- and D-layers.

The existence of different percolation thresholds for localized perturbations in interdependent networks may reveal itself as a universal principle for balancing sensitivity and robustness in complex systems. The application of these concepts to a wide range of real-life systems is required to make progress in this direction.

## Methods

**Network**. The integrative network of *E. coli* has been assembled using the EcoCyc database[35, 36] (version 18.5), which offers both the data about metabolic processes and (gene) regulatory events incorporated from the RegulonDB[66] (release 8.7). The extensive metadata allows for the assignment of the vertices to one of the three functional domains. Details of this process and a detailed characterization of the resulting model are described in (A.G., D.F.K., S.B., and M.-T.H., manuscript in preparation). The corresponding graph representation consists of 10,383 vertices and 24,150 directed edges.

Since we are interested in the propagation of a signal between the domains, in the following we will refer to the domains of the source and target vertices of the edge $e_k = \left( v_s^{(k)}, v_t^{(k)} \right)$ as source domain, SD, and target domain, TD, respectively.

The metadata can be used to assign properties to the vertices and edges of the graph beyond the domain structure, some of which are used in the following analysis, namely in the compilation of the percolation model and the randomization schemes.

We distinguish between biological categories of edges (capturing the diverse biological roles of the edges) and the logical categories (determining the rules of the percolation process). According to their biological role in the system, both vertices and edges are assigned to a biological category; we abbreviate the biological category of a vertex as BCV and the biological category of an edge as BCE (for details see Supplementary Note 2). Each of the eight BCEs can then be mapped uniquely to one of only three LCE,

$$e_k^{LCE} \in \{C, D, R\}.$$

**Percolation model**. In the dynamical rules describing the propagation of an initial perturbation in the network, we distinguish between the different roles a given edge has in the update of a target vertex in terms of the LCE.

In our percolation model, every vertex is assigned a Boolean state variable $\sigma \in \{0, 1\}$; since we intend to mimic the propagation of a perturbation (rather than simulate a trajectory of actual biological states), we identify the state 1 with 'not yet affected by the perturbation', while the state 0 corresponds to 'affected by the perturbation'. We stress that the trajectory $\sigma(t)$ does not correspond to the time evolution of the abundance of gene products and metabolic compounds, but the rules have been chosen such that the final set of affected vertices provides an estimate of all vertices potentially being affected by the initial perturbation. A vertex not in this set is topologically very unlikely of being affected by the perturbation at hand (given the biological processes contained in our model).

A stepwise update can now be defined for vertex $i$ with in-neighbors $\Gamma_i^-$ in order to study the spreading of perturbations through the system by initially switching off a fraction $q$ of vertices:

$$\sigma_i(t+1) = f_i\big(\sigma_j(t)|\sigma_j \in \Gamma_i^-\big) \qquad (1)$$

$$
f_i =
\begin{cases}
1 & \text{if } \left[\sum_j c_{ij}(1-\sigma_j) = 0\right] \quad \wedge \\
& \quad \left[\sum_j d_{ij}\sigma_j > 0 \vee \sum_j d_{ij} = 0\right] \quad \wedge \\
& \quad \left[\sum_j |r_{ij}|(1-\sigma_j) = 0\right] \\
0 & \text{otherwise}
\end{cases}
\qquad (2)
$$

where $c_{ij}$ is 1 if $v_j$ is connected to $v_i$ via a C-edge and 0 otherwise; $d_{ij}$ and $r_{ij}$ are defined analogously.

Thus, a vertex will be considered unaffected by the perturbation if none of its in-neighbors connected via either a C- or an R-edge have failed (regardless of the type of regulatory interaction, activation or inhibition), and at least one of its in-neighbors connected via a D-edge is still intact. With an additional rule, it is ensured that an initially switched off vertex stays off. The choice of the update rules ensures that the unperturbed system state is conserved under the dynamics: **f(1) = 1**.

**Network response to localized perturbation analysis**. Due to the functional three-domain partition of our *E. coli* gene regulatory and metabolic network reconstruction, we have the possibility to classify perturbations not only according to their size, but also with respect to their localization in one of the domains comprising the full interdependent system, thereby enabling us to address the balance of sensitivity and robustness of the interdependent network of gene regulation and metabolism.

Here we introduce the concept of network response to localized perturbations analysis. This analysis will reveal that perturbations in gene regulation affect the system in a dramatically different way than perturbations in metabolism. Thus, we study the response to localized perturbations, which we denote by Per($X$, $q$), where $X$ is the domain, in which the perturbation is localized ($X \in \{R, I, M, T\}$, with T representing the total network $G$, i.e., the case of non-localized perturbations). The perturbation size $q = 1 - p$ is measured in fractions of the total network size $N = |G|$. Hence, Per($M$, 0.1) is a perturbation in the metabolic domain with (on average) $0.1|G|$ vertices initially affected. Note that sizes $q$ of such localized perturbations are limited by the domain sizes, e.g., $q|G| < |G_R|$ for a perturbation in the gene regulatory domain.

After the initial perturbation of a fraction $q$ of the vertices in the domain $X$ the stepwise dynamics described above will lead to the deactivation of further vertices and run into a frozen state $\sigma_\infty$, in which only a fraction $A(X, q)$ of the vertices are unaffected by the dynamics (i.e., are still in state 1). In addition to being directly affected by failing neighbors, in the process of network fragmentation vertices may also become parts of small components disconnected from the network's core, and could in this sense be considered non-functional; we therefore also monitor the

relative sizes of the largest (weakly) connected component in the frozen state, $B(X, q)$, for different initial perturbation sizes.

In the limit of infinite system size, we could expect a direct investigation of $B(X, q)$ as a function of $q$ to yield the critical perturbation size $q_c = 1 - p_c$ at which $B$ vanishes. In our finite system, however, we have to estimate $q_c$; following Radicchi and Castellano[39], and Radicchi[67], we measure the fluctuation of $B(X, q)$ which serves as our order parameter and look for the peak position of the susceptibility

$$\tilde{\chi}(X, q) = \langle B_\infty^2 \rangle - \langle B_\infty \rangle^2 \qquad (3)$$

as a function of parameter $q$ in order to estimate the transition point from the finite system data. Supplementary Fig. 6 schematically illustrates the analysis (also see Supplementary Note 1).

**Randomization schemes**. In order to interpret the actual responses of a given network to perturbations, one usually contrasts them to those of suitably randomized versions of the network at hand. Thereby, the often dominant effect of the vertex degree distribution of a network can be accounted for and the effects of higher-order topological features that shape the response of the network to perturbations can be studied systematically.

The same is true for the localized perturbation response analysis introduced here. In fact, due to the substantially larger number of edges from gene regulation to metabolism (both, directly and via the interface component of the interdependent network) than from metabolism to gene regulation we can already expect the response to such localized perturbations to vary.

Here we employ a sequence of ever more stringent randomization schemes to generate sets of randomized networks serving as null models for the localized perturbation response analysis. In all of the four schemes, the edge-switching procedure introduced by Maslov and Sneppen[68] is employed which conserves the in- and out-degrees of all vertices.

Our most flexible randomization scheme (DOMAIN) only considers the domains of the source and target vertices of an edge (SD and TD): only pairs of edges are flipped which share both, the source and the target domain (e.g., both link a vertex in the metabolic domain to a vertex in the interface). The remaining three randomization schemes all add an additional constraint. The DOMAIN_LCE randomization further requires the edges to be of the same logical categories of an edge (i.e., C, D, or R), while the DOMAIN_BCV scheme only switches edges whose target vertices also share the same biological category of a vertex, BCV. The strictest randomization, DOMAIN_BCE, finally, only considers edges with, additionally, the same biological category of an edge, BCE. A tabular overview of the four schemes is given in Supplementary Table 1.

**Data availability**. The data that support the findings of this study are available from the authors upon reasonable request.

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

## Acknowledgements

S.B. and M.-T.H. acknowledge the support of Deutsche Forschungsgemeinschaft (DFG), grants BO 1242/6 and HU 937/9.

## Author contributions

M.-T.H. and S.B. designed and supervised the study; D.K. and A.G. performed the reconstruction, simulations and analyses; and D.K., A.G., S.B. and M.-T.H. wrote the manuscript.

## Additional information

**Competing interests:** The authors declare no competing financial interests.

