## [Peer Review File · Nature Communications]

Reviewers' comments:

Reviewer #1 (Remarks to the Author):

The authors address the problem of the spreading of signals in a biologically-motivated interdependent network, describing the interactions between the metabolic and the gene-regulatory system in a biological cell. After reconstructing the net of the interactions in terms of a complete, directed graph composed by three networks, the authors address the spreading of damage via a percolation process, assessing hence the robustness of the system as a whole. This manuscript presents an innovative and important application of interdependent network theory to biological systems. On the theoretical front, it presents a new type of network measurement by considering the "effects" of a perturbation localized in a given domain. The study introduces a novel concept, previously unknown in the study of interdependent system, namely the network response to localized attacks, which are initially seeded selectively in a chosen network module. This will likely prove useful in numerous applications. On the applied front, they demonstrate how the simultaneous robustness and sensitivity of the protein network can be understood by separating the metabolic links (which are robust) from the gene regulatory links (which are sensitive). Overall the paper is innovative with several original results, well-written and sure to be appreciated and used by many researchers working in network science as well as systems biology, since it bridges between these fields. This paper potentially deserves publishing in a high impact journal as Nature Communication, with few major and minor adjustments. Several major points still require some elucidation though. First, the data acquisition and processing (upon which the entire paper is based) is not explained here but rather in another manuscript. Without seeing the other, I cannot recommend publication of this manuscript. That is because key decisions about how different biological metadata is translated into network properties are made there yet changes in that translation process might have major impacts on the results here.

Second, in Sec.III and Sec.VII, the authors point out in several places the limitation in the applicability of the mathematical concepts of interdependent networks to real-data, requiring for "substantial adjustments". In III they explicitly refer to the limitations of the "alternating percolation and dependency steps" (considered in modeling the dynamics of cascades in interdependent percolation), and in VII they write that "the notion of dependency links vs. connectivity links is no longer applicable". For better understanding these claims, a more clear description of the step wise dynamics has to be included, exemplifying it schematically (e.g., for the first three steps) as done in the original paper by Buldyrev et al. (Nature 464, 2010, pg.1026, Fig.2), and comparing then with the usual ("alternating") process considered so far. More specific theoretical points in the text that require some clarification: 1) On the top right of p. 4, at the end of section III the authors write: "in our directed model both, connectivity and dependency links are evaluated in every time step and (apart from nodes failing due to dependency) only fully decoupled vertices cause further dependency failures." It is not clear how this differs from the standard approach. Is the only difference that the survival criterion is not limited to the largest component only (see e.g., Yuan et al, arXiv:1605.04217) ? What role does the change in the evaluation method play? Why is this change necessary or preferable? 2) On the middle right of p. 6 paragraph beginning "Here we have" the authors write: "As a consequence, the notion of dependency links vs. connectivity links is no longer applicable." It is not clear what the intention is. This is a strong sounding conclusion and yet it's not clear at all how this sentence follows the preceding sentences in the paragraph. Is it because there are many other roles that can be played as

signified by the "R" links here? 3) In Fig. 2 "trajectories" are shown but it is not clear what they mean. What is the y axis? Order parameter A? B? How are the colors and the dashed line to be interpreted? How can the value go up and down? In standard interdependent network cascades this is not possible, and it is not clear how such a scenario can arise from the description of the model here and its differences with previous work. The caption and related discussion in the text leave the answer to these questions unclear. Minor issues. 1) In the abstract, second paragraph, it should be mentioned explicitly the word "percolation" when writing about "the theoretical framework of interdependent networks", and the same has to be done in the introduction. 2) In

Sec. II, the index i is used for enumerating vertices, whilst in Sec. I is used for links, leading to an ambiguous enumeration. 3) Sec. III the first sentence is missing a dot and contains the statement "response to average perturbations of a given size": it is unclear what the authors mean for "average perturbation". Soon after the susceptibility related to the giant component is introduced, but the average there considered is not specified. 4) In Sec. VI it is written that cascades are "seeded in the...", more appropriately they are "ignited in the..." (or better terms) as a random failure might also not lead to any cascade. 5) In Sec. VI, second indentation, it is written that "This effect can be seen both in [...] as well as in the domain which shows the largest change with respect to the previous time step, [...]"; the definition of this measure is unclear and should be better explained. 6) In Sec. VI, second-last indentation, it is written "the average size of the [...] under a perturbation"; the authors may want to add "of given size q " before ending the previous sentence. 7) Sec. VII, eighth indentation, two equivalent definitions of interdependent networks are considered: some proper references should be added consistently with the claims made. 8) In Bibliography. We would like to point out that Ref. 23 should be inserted before Ref. 18, being the first research towards the interdependent network formalism. Moreover, before Ref. 39, the previous paper by Majdandzic et al. (Nature Physics, 10, 34-38, 2014) could be also referred, being the first one introducing the concept of network recovery. In general, I think that this work is an important and timely contribution to build an important bridge between network science and systems biology. Once the details of the data collection and processing are clarified as well as the relation to previous theoretical work is better explained, I would gladly recommend the manuscript to be published in Nature Communication.

Reviewer #2 (Remarks to the Author):

Klosik et al. manuscript presents a study of robustness of a multilayer molecular network consisting on regulatory and metabolic interactions.

Multilayer networks is a rapidly expanding subject of network theory that is very promising for developing a more comprehensive understanding of molecular interaction in the cell. For instance network medicine requires the study of the interactome that is essentially a multilayer network. Therefore the research question addressed by this paper has the potential to attract significant interest.

In network theory generalized percolation transitions are extensively studied and there is a general debate about the implication of multiplexity on the robustness properties of the networks.

For instance interdependencies have been claimed to make multilayer networks rather fragile, but this poses important questions when one aims at explaining the occurrence of multiplexity in biological networks. This question has been the focus of important previous publications [1-3] that need to be correctly cited here.

Specifically ref. [3] is particularly relevant for this paper because there for the first time a complete theoretical treatment of "redundant interdependencies" and their role in improving the robustness of multilayer network is given. This redundant interdependencies are practically the disjunct interactions presented in this paper.

The present paper somehow complements these studies with the analysis of a carefully collected molecular biology dataset including both interdependencies, and redundant interdependencies (called here conjunct and disjunct interactions). The result is that biological networks respond differently to perturbations localized in different region of the multilayer network and that overall display a robust behaviour when compare with suitable null models.

Overall I think that this study could become suitable for publication to Nature Communication when the Authors acknowledge relevant previous research on robustness of multilayer networks and moderate somehow their tone addressing the following points:

- a) Appropriate reference to publications [1,2] raising the question of robustness in biological multilayer networks for the first time should be given.
- b) The sentences at the end of page 4 on the applicability of mathematical concepts in real world

network should be removed. It seems to me that the Authors are making an argument against the same approach that they take. Is not this percolation study just a mathematical abstraction of the biological problem? Moreover novel theoretical techniques such as message passing go well beyond any distinction between analytically solvable models and simulations. Finally actually the insights gained by analytical models of redundant interdependencies such as the ones obtained in [3] are very relevant to this study.

c) More insights into the mathematical developments of percolation in multilayer networks should be given.

First of all [3] should be acknowledged in length and the relation between the disjoint interaction studied in this paper and the redundant interactions studied in [3] should be clearly stated.

Secondly the Authors should be aware of the distinction of "dynamical" "feedforward" algorithms and the generalized giant components in multilayer networks. For instance they should read the paper [4] clarifying the distinction between "directed percolation of multilayer networks" (studied for instance in paper Ref. [22] of this manuscript and previously already in [6]) and "percolation of multilayer networks".

Thirdly when citing percolation on network of networks I suggest the Authors to consider citing [5] where an interesting percolation transition is observed to take place in different layers for different values of the initial damage similarly to the present simulation results.

[1] Reis, Saulo DS, et al. "Avoiding catastrophic failure in correlated networks of networks." *Nature Physics* 10.10 (2014): 762-767.

[2] Bianconi, Ginestra. "Multilayer networks: Dangerous liaisons?." *Nature Physics* 10.10 (2014): 712-714.

[3] Radicchi, Filippo, and Ginestra Bianconi. "Redundant interdependencies boost the robustness of multiplex networks." *Physical Review X* 7.1 (2017): 011013.

[4] Cellai, Davide, Sergey N. Dorogovtsev, and Ginestra Bianconi. "Message passing theory for percolation models on multiplex networks with link overlap." *Physical Review E* 94.3 (2016): 032301.

[5] Bianconi, Ginestra, and Sergey N. Dorogovtsev. "Multiple percolation transitions in a configuration model of a network of networks." *Physical Review E* 89.6 (2014): 062814.

[6] Cellai, Davide, et al. "Percolation in multiplex networks with overlap." *Physical Review E* 88.5 (2013): 052811.

Reviewer #1:

The authors address the problem of the spreading of signals in a biologically-motivated interdependent network, describing the interactions between the metabolic and the gene-regulatory system in a biological cell. After reconstructing the net of the interactions in terms of a complete, directed graph composed by three networks, the authors address the spreading of damage via a percolation process, assessing hence the robustness of the system as a whole. This manuscript presents an innovative and important application of interdependent network theory to biological systems. On the theoretical front, it presents a new type of network measurement by considering the "effects" of a perturbation localized in a given domain. The study introduces a novel concept, previously unknown in the study of interdependent system, namely the network response to localized attacks, which are initially seeded selectively in a chosen network module. This will likely prove useful in numerous applications. On the applied front, they demonstrate how the simultaneous robustness and sensitivity of the protein network can be understood by separating the metabolic links (which are robust) from the gene regulatory links (which are sensitive).

Overall the paper is innovative with several original results, well-written and sure to be appreciated and used by many researchers working in network science as well as systems biology, since it bridges between these fields. This paper potentially deserves publishing in a high impact journal as Nature Communication, with few major and minor adjustments.

We thank the reviewer for his/her commendatory lines regarding our work. In the response to the reviewer's comments and the revised manuscript, we clarify and address the remaining concerns.

Several major points still require some elucidation though.

First, the data acquisition and processing (upon which the entire paper is based) is not explained here but rather in another manuscript. Without seeing the other, I cannot recommend publication of this manuscript. That is because key decisions about how different biological metadata is translated into network properties are made there yet changes in that translation process might have major impacts on the results here.

We welcome the reviewer's request to provide the associated manuscript describing the reconstruction procedure. As part of the resubmission, we make the relevant extracts of our manuscript, which is currently being prepared for submission, confidentially available for the reviewer. We hope these extracts convey the two important points:

- **The network construction process from data is biologically and technically sound.**
- **The main feature of the resulting network representation, that is its prominent three-domain organization is a meaningful partition. This is shown by comparing appropriate measures of modularity for this and alternative partitions.**

Second, in Sec.III and Sec.VII, the authors point out in several places the limitation in the applicability of the mathematical concepts of interdependent networks to real-data, requiring for "substantial adjustments". In III they explicitly refer to the limitations of the "alternating percolation and dependency steps" (considered in modeling the dynamics of cascades in interdependent percolation), and in VII they write that "the notion of dependency links vs. connectivity links is no longer applicable". For better understanding these claims, a more clear description of the step wise dynamics has to be included, exemplifying it schematically (e.g., for the first three steps) as done in the original paper by Buldyrev et al. (Nature 464, 2010, pg.1026, Fig.2), and comparing then with the usual ("alternating") process considered so far.

Such a scheme is indeed helpful, not only to emphasize the conceptual difference to the percolation model used in Buldyrev et al. [5] but also to make the different node and link attributes, which are present in our network, more accessible. We added a schematic representation of a small network showing the cascading failure triggered by a initial single-vertex perturbation as a new Fig. 2.

Additionally, a more detailed illustration of the process has been added to the Supplementary Materials as Figure S2.

Furthermore, we agree that our statements masked the fact that the similarities between our percolation model and the approach introduced in Buldyrev et al. [5] and subsequent works are stronger than the differences. Therefore we deleted the sentence containing the terms "substantial adjustments" and instead added the following paragraph:

Clear conceptual categories, such as the distinction between dependency edges and connectivity edges have been instrumental in gaining theoretical insight into the properties of interdependent systems. Similarly, in the theoretical work on multilayer networks (see, e.g., De Domenico *et al.* [10, *in manuscript: 55*], Boccaletti *et al.* [4, *in manuscript: 56*]), it is assumed that the assignment of vertices and edges to layers is defined *a priori*.

It is worth mentioning, however, that in order to adapt these frameworks to the situation at hand several of these categories require adaptation. We envision that for many real-life applications a certain diversity of vertex and edge attributes will be required and the multilayer structure will rather emerge from the arrangement of such different edge types among the different vertex types in the system.

Relatedly, we deleted the sentence in the Outlook section saying "the notion of dependency links vs. connectivity links is no longer applicable" and instead added the following paragraph:

As a consequence, our investigation required an adjustment of the original percolation scheme described in Buldyrev *et al.* [5, *in manuscript: 17*] to the vertex and edge classes employed here. We expect that application of the notion of interdependent networks to real-life systems is likely to require adjustment of the percolation model, even though the main features of the original theory remain fully applicable.

More specific theoretical points in the text that require some clarification:

- 1) On the top right of p. 4, at the end of section III the authors write: "in our directed model both, connectivity and dependency links are evaluated in every time step and (apart from nodes failing due to dependency) only fully decoupled vertices cause further dependency failures." It is not clear how this differs from the standard approach. Is the only difference that the survival criterion is not limited to the largest component only (see e.g., Yuan et al, arXiv:1605.04217)? What role does the change in the evaluation method play? Why is this change necessary or preferable?

Our criterion for the impact of a perturbation is the total number of affected nodes after perturbation is allowed to propagate through the interdependent network according to the set of update rules derived from biological functions. Fluctuations in the size of the largest component then serve as an indicator for the loss-of-function threshold in this system. We thus do not assume that nodes isolated from the largest component can function autonomously, as discussed in Yuan et al. [19].

Since the dependency links in the system studied here do not coincide with the inter-domain links there is no direct equivalent of the mutually connected component. Thus, we have to adapt the rule that being decoupled from the largest mutually connected component leads to failure, and do so via the update rule for 'disjunct' links: a node (that does not already fail due to a failing dependency link) only fails if all connectivity links fail. Finally, in the steady-state, we only consider the largest connected component.

This statement has also been added to the revised version of the manuscript at the end of chapter II A.

With the explicitly alternating 'percolation' and 'dependency' steps in typical computational models of the kind originally introduced in Buldyrev *et al.* [5, *in manuscript: 17*] in mind we would like to point out that in our directed model edges of all types are updated in every time step. If the dependencies of a vertex are fulfilled (here: only intact C- and R-neighbours) it will only fail once it loses all connectivity (here: via D-edges), irrespective of the current component structure, which is only considered after the dynamics is frozen.

- 2) On the middle right of p. 6 paragraph beginning "Here we have" the authors write: "As a consequence, the notion of dependency links vs. connectivity links is no longer applicable." It is not clear what the intention is. This is a strong sounding conclusion and yet it's not clear at all how this sentence follows the preceding sentences in the paragraph. Is it because there are many other roles that can be played as signified by the "R" links here?

We realize now that the statement conveys the wrong impression and have modified this paragraph accordingly (see our reply above).

- 3) In Fig. 2 "trajectories" are shown but it is not clear what they mean. What is the y axis? Order parameter A? B? How are the colors and the dashed line to be interpreted? How can the value go up and down? In standard interdependent network cascades this is not possible, and it is not clear how such a scenario can arise from the description of the model here and its differences with previous work. The caption and related discussion in the text leave the answer to these questions unclear.

We added the following explanation to the figure caption (*please note that due to the introduction of a further figure the old Fig. 2 is now Fig. 3.*)

In each panel the vertex states are plotted against the update steps according to equations (1) and (2). The vertices are grouped according to the domain affiliation, starting from gene regulatory domain (top, green) via protein interface domain (middle, blue) towards metabolic domain (bottom, red). Within each domain they are arranged with respect to the update step, in which they were affected in the respective run (vertex ordering within the domains is thus different from run to run). Affected vertices are shown in the color of their domain, unaffected vertices are shown last for each run and in white. The horizontal dashed lines denote the domain boundaries and the black square indicates the domain with the largest number of newly affected vertices in each step. Connections between black squares are only shown to guide the eye.

Minor issues.

- 1) In the abstract, second paragraph, it should be mentioned explicitly the word "percolation" when writing about "the theoretical framework of interdependent networks", and the same has to be done in the introduction.

We added the term 'percolation' to the abstract; we also included 'percolation' in the introduction.

- 2) In Sec. II, the index i is used for enumerating vertices, whilst in Sec.I is used for links, leading to an ambiguous enumeration.

We now use k as an index enumerating links.

- 3) Sec. III the first sentence is missing a dot and contains the statement "response to average perturbations of a given size": it is unclear what the authors mean for "average perturbation". Soon after the susceptibility related to the giant component is introduced, but the average there considered is not specified.

We corrected the typo and deleted the word "average" (which is indeed misleading: In Figure 4 the susceptibility is shown, which is derived from the distribution of responses, but not the average, while Figure 5 indeed shows the average number of unaffected nodes).

- 4) In Sec. VI it is written that cascades are "seeded in the...", more appropriately they are "ignited in the..." (or better terms) as a random failure might also not lead to any cascade.

At the four places in our manuscript, where we talk about cascades "seeded in" a domain, we now substituted this expression by "initiated in" a domain.

- 5) In Sec. VI, second indentation, it is written that "This effect can be seen both in [...] as well as in the domain which shows the largest change with respect to the previous time step, [...]"; the definition of this measure is unclear and should be better explained.

This should now be clearer due to the extended caption of Figure 3 (see above). Additionally we modified the text "which shows the largest change with respect to the previous time step" in the following way: "which shows the largest change with respect to the previous time step (largest set of newly affected nodes)".

- 6) In Sec. VI, second-last indentation, it is written "the average size of the [...] under a perturbation"; the authors may want to add "of given size q " before ending the previous sentence.

We added this.

- 7) Sec. VII, eighth indentation, two equivalent definitions of interdependent networks are considered: some proper references should be added consistently with the claims made.

Based on previous work in Neuroscience [e.g., 12] and Systems Biology [e.g., 14] we perceived a gap between the two definitions. In the light of a publication Morone et al. [15], which appeared just a few weeks ago, we see how the two views can be related. We therefore have reformulated this paragraph. It now reads:

In the vast majority of theoretical investigations interdependent networks are defined via a distinction between dependency edges and connectivity-representing edges [5, 8, 16, in manuscript: 17,19,57]. Often, in specific applications this distinction of connectivity edges and dependency edges is not immediately clear (see, e.g., examples from neuroscience [12, in manuscript: 58] and systems biology [14, in manuscript: 59]). In Morone et al. [15, in manuscript: 60] an elegant way of mapping such real-life systems to the formal description of connectivity edges and dependency edges has been worked out for the case of brain networks by emphasizing the different roles of intramodule and intermodule edges.

8) In Bibliography. We would like to point out that Ref.23 should be inserted before Ref.18, being the first research towards the interdependent network formalism. Moreover, before Ref.39, the previous paper by Majdandzic et al. (Nature Physics, 10, 34-38, 2014) could be also referred, being the first one introducing the concept of network recovery.

We adjusted the references accordingly. Especially, the former reference [23] is now reference [17] (while [18] has not been moved). Majdandzic et al. (Nature Physics, 10, 34-38, 2014) is now cited as [47], before the former Ref.39 which is now [48].

In general, I think that this work is an important and timely contribution to build an important bridge between network science and systems biology. Once the details of the data collection and processing are clarified as well as the relation to previous theoretical work is better explained, I would gladly recommend the manuscript to be published in nature communication.

Reviewer #2:

Klosik et al. manuscript presents a study of robustness of a multilayer molecular network consisting on regulatory and metabolic interactions. Multilayer networks is a rapidly expanding subject of network theory that is very promising for developing a more comprehensive understanding of molecular interaction in the cell. For instance network medicine requires the study of the interactome that is essentially a multilayer network. Therefore the research question addressed by this paper has the potential to attract significant interest. In network theory generalized percolation transitions are extensively studied and there is a general debate about the implication of multiplexity on the robustness properties of the networks. For instance interdependencies have been claimed to make multilayer networks rather fragile, but this poses important questions when one aims at explaining the occurrence of multiplexity in biological networks. This question has been the focus of important previous publications [1-3 *here*: [1, 17, 18]] that need to be correctly cited here.

Even though our focus is *not* on general aspects of robustness in multilayer systems or on the evolution of multilayer networks, we fully agree with the reviewer that the corresponding literature should be cited in our manuscript which we now do. We particularly thank the reviewer for pointing out the very recent reference [17], which we now discuss in detail.

Specifically ref. [3 *here*: [17]] is particularly relevant for this paper because there for the first time a complete theoretical treatment of "redundant interdependencies" and their role in improving the robustness of multilayer network is given. This redundant interdependencies are practically the disjunct interactions presented in this paper.

We agree with the assessment of the reviewer that potentially our disjunct links serve as 'redundant interdependencies', as defined in Radicchi and Bianconi [17]. We therefore added the following paragraph:

As outlined in the Methods section the conjunct and disjunct edge properties in our study have been derived from the underlying biological mechanisms. The disjunct edges may be examples of the redundant interdependencies, which have been discussed in Radicchi and Bianconi [17, *in manuscript*: 31] as a possible foundation of the robustness of multilayer (specifically, multiplex) networks.

The present paper somehow complements these studies with the analysis of a carefully collected molecular biology dataset including both interdependencies, and redundant interdependencies (called here conjunct and disjunct interactions). The result is that biological networks respond differently to perturbations localized in different region of the multilayer network and that overall display a robust behaviour when compare with suitable null models. Overall I think that this study could become suitable for publication to Nature Communication when the Authors acknowledge relevant previous research on robustness of multilayer networks and moderate somehow their tone addressing the following points:

- a) Appropriate reference to publications [1,2 *here*: [1, 18]] raising the question of robustness in biological multilayer networks for the first time should be given.

We added the following paragraph addressing the broader context of robustness in multilayer networks

Even though the framework of multiplex networks does not strictly apply to the example of a multilayer network discussed here, as there is no one-to-one mapping from vertices in one layer to vertices in the other layers, several theoretical findings on multiplex networks nevertheless can help us to put the results of our investigation in a broader context. It has been recognized in a previous study [17,

in manuscript: 31] that understanding the robustness of multilayer networks may require adapting the standard percolation model, where a vertex is switched off, if at least one of the interdependent vertices has been switched off as well. This percolation model is at the core of many of the investigations of catastrophic failures of interdependent networks, e.g., Buldyrev et al. [5, *in manuscript: 17*]. In this scenario, a larger number of layers will typically lead to more vulnerable networks. The interesting question, how an increasing number of layers might actually enhance the robustness of the system, has been addressed in Radicchi and Bianconi [17, *in manuscript: 31*]. In our case the cascading failure, and hence the percolation model, is determined by the Boolean update rules in equations (1) and (2) which have been derived from the underlying biological processes as described in the corresponding Results and Methods sections (and in more detail in the Supplementary Information).

This biologically motivated percolation model used here is comparable to the model of percolation in multiplex networks introduced in Radicchi and Bianconi [17, *in manuscript: 31*]. Even though intracellular molecular network systems, as the one studied here, tend to not adhere to the formal requirement of a one-to-one mapping of vertices between different layers, it is likely that the main finding from [17, *in manuscript: 31*] – that an addition of layers enhances systemic robustness by providing redundant interdependencies – may still hold. It would be interesting to further explore in what way the multilayer structure of intracellular molecular networks indeed enhances their robustness in this way.

Furthermore, we acknowledge the work from Reis et al. [18] in the following newly added paragraph:

In Reis *et al.* [18, *in manuscript: 24*] the question of robustness in multilayer biological networks has been raised for the first time (see also Bianconi [1, *in manuscript: 25*]). Specifically, it has been shown how specific correlations of the intra- and interlayer connections reduce cascading failures and thus provide a robust multilayer network architecture.

- b) The sentences at the end of page 4 on the applicability of mathematical concepts in real world network should be removed. It seems to me that the Authors are making an argument against the same approach that they take. Is not this percolation study just a mathematical abstraction of the biological problem? Moreover novel theoretical techniques such as message passing go well beyond any distinction between analytically solvable models and simulations.

We already substantially modified these paragraphs in response to suggestions by Reviewer #1. In order to avoid the misconception that we insinuate a lack of applicability of the vast bulk of theoretical work on multilayer networks, we also added the following paragraph:

Relevant progress in the application of concepts of multilayer networks has been made, for instance, in transportation infrastructures [11, *in manuscript: 26*] and brain networks [18, *in manuscript: 24*]. In such applications the discovery of new mechanisms went along with advances in the theoretical foundation in exploring dynamical processes in multilayer networks, for example diffusion processes [13, *in manuscript: 27*], spreading processes [9, *in manuscript: 28*] and message passing [3, 6, 17, *in manuscript: 29-31*].

Finally actually the insights gained by analytical models of redundant interdependencies such as the ones obtained in [3 *here: [17]*] are very relevant to this study.

In addition to the new paragraphs listed above, we also now point out a possible link between the results from [18] and our findings:

Also, it would be interesting to investigate in more detail, whether the pattern of dependency edges, which in Reis *et al.* [18, *in manuscript: 24*] has been associated with robustness of interdependent networks, are more prominent in the metabolism-to-regulation direction of our system than in the regulation-to-metabolism direction.

c) More insights into the mathematical developments of percolation in multilayer networks should be given.

- First of all [3 *here: [17]*] should be acknowledged in length and the relation between the disjunct interaction studied in this paper and the redundant interactions studied in [3 *here: [17]*] should be clearly stated.

This point has now been addressed (see the new text paragraphs listed above).

- Secondly the Authors should be aware of the distinction of "dynamical" "feedforward" algorithms and the generalized giant components in multilayer networks. For instance they should read the paper [4 *here: [6]*] clarifying the distinction between "directed percolation of multilayer networks" (studied for instance in paper Ref. [22 (*now 23*)] of this manuscript and previously already in [6 *here: [7]*]) and "percolation of multilayer networks".

We would like to emphasize that here we study one specific biological network under a percolation model motivated by the biological function. However, in acknowledgement of the points put forward by the reviewer, we added the following paragraph:

Important contributions to the theoretical understanding of percolation phenomena in multilayered structures come from studying variants of message passing algorithms on multiplex networks. Cellai *et al.* [6, *in manuscript: 29*], for example, show how the presence of link overlap between layers can be considered in terms of two similar, but conceptually different, processes and analyse the behaviour of the corresponding order parameters.

We added this reference to the citations in Section III, where we discuss percolation on networks of networks, as well as to the paragraph listed in the previous response (letter b).

- Thirdly when citing percolation on network of networks I suggest the Authors to consider citing [5 *here: [2]*] where an interesting percolation transition is observed to take place in different layers for different values of the initial damage similarly to the present simulation results.

We agree that [2, *in manuscript: 45*] certainly should be cited in the 'network of networks'-context and added a reference in chapter II A.

References

- [1] Bianconi, G. (2014). Multilayer networks: Dangerous liaisons? *Nature Physics*, 10(10):712–714.
- [2] Bianconi, G. and Dorogovtsev, S. N. (2014). Multiple percolation transitions in a configuration model of a network of networks. *Physical Review E*, 89(6):062814.

- [3] Bianconi, G. and Radicchi, F. (2016). Percolation in real multiplex networks. *Physical Review E*, 94(6):060301.
- [4] Boccaletti, S., Bianconi, G., Criado, R., Del Genio, C. I., Gómez-Gardenes, J., Romance, M., Sendina-Nadal, I., Wang, Z., and Zanin, M. (2014). The structure and dynamics of multilayer networks. *Physics Reports*, 544(1):1–122.
- [5] Buldyrev, S. V., Parshani, R., Paul, G., Stanley, H. E., and Havlin, S. (2010). Catastrophic cascade of failures in interdependent networks. *Nature*, 464(7291):1025–1028.
- [6] Cellai, D., Dorogovtsev, S. N., and Bianconi, G. (2016). Message passing theory for percolation models on multiplex networks with link overlap. *Physical Review E*, 94(3):032301.
- [7] Cellai, D., López, E., Zhou, J., Gleeson, J. P., and Bianconi, G. (2013). Percolation in multiplex networks with overlap. *Physical Review E*, 88(5):052811.
- [8] Danziger, M. M., Shekhtman, L. M., Bashan, A., Berezin, Y., and Havlin, S. (2016). Vulnerability of interdependent networks and networks of networks. In *Interconnected Networks*, pages 79–99. Springer.
- [9] De Domenico, M., Granell, C., Porter, M. A., and Arenas, A. (2016). The physics of spreading processes in multilayer networks. *Nature Physics*.
- [10] De Domenico, M., Solé-Ribalta, A., Cozzo, E., Kivela, M., Moreno, Y., Porter, M. A., Gómez, S., and Arenas, A. (2013). Mathematical formulation of multilayer networks. *Physical Review X*, 3(4):041022.
- [11] De Domenico, M., Solé-Ribalta, A., Gómez, S., and Arenas, A. (2014). Navigability of interconnected networks under random failures. *Proceedings of the National Academy of Sciences*, 111(23):8351–8356.
- [12] Deco, G., Jirsa, V. K., and McIntosh, A. R. (2011). Emerging concepts for the dynamical organization of resting-state activity in the brain. *Nat Rev Neurosci*, 12(1):43–56.
- [13] Gomez, S., Diaz-Guilera, A., Gomez-Gardenes, J., Perez-Vicente, C. J., Moreno, Y., and Arenas, A. (2013). Diffusion dynamics on multiplex networks. *Physical review letters*, 110(2):028701.
- [14] Menche, J., Sharma, A., Kitsak, M., Ghiassian, S. D., Vidal, M., Loscalzo, J., and Barabási, A.-L. (2015). Uncovering disease-disease relationships through the incomplete interactome. *Science*, 347(6224):1257601.
- [15] Morone, F., Roth, K., Min, B., Stanley, H. E., and Makse, H. A. (2017). Model of brain activation predicts the neural collective influence map of the brain. *Proceedings of the National Academy of Sciences*, 114(15):3849–3854.
- [16] Parshani, R., Buldyrev, S. V., and Havlin, S. (2011). Critical effect of dependency groups on the function of networks. *Proc. Natl Acad. Sci. USA*, 108(3):1007–1010.
- [17] Radicchi, F. and Bianconi, G. (2017). Redundant interdependencies boost the robustness of multiplex networks. *Physical Review X*, 7(1):011013.
- [18] Reis, S. D., Hu, Y., Babino, A., Andrade Jr, J. S., Canals, S., Sigman, M., and Makse, H. A. (2014). Avoiding catastrophic failure in correlated networks of networks. *Nature Physics*, 10(10):762–767.
- [19] Yuan, X., Hu, Y., Stanley, H. E., and Havlin, S. (2017). Eradicating catastrophic collapse in interdependent networks via reinforced nodes. *Proceedings of the National Academy of Sciences*, page 201621369.

REVIEWERS' COMMENTS:

Reviewer #1 (Remarks to the Author):

The authors revised the manuscript properly according to my comments and the comments of the second referee. I find the current version of the manuscript of great interest and recommend its publication in the present form.

Reviewer #2 (Remarks to the Author):

The Authors have now addressed all my comments and I judge the paper suitable for Nature Communication.